# Cultivation of *Arthrospira platensis* in Veterinary Hospital Wastewater Enhances Pigment Production and Reduces Antibiotic Resistance Genes

**DOI:** 10.3390/biology14101396

**Published:** 2025-10-12

**Authors:** Authen Promariya, Sekbunkorn Treenarat, Nattaphong Akrimajirachoote, Wanat Sricharern, Wuttinun Raksajit

**Affiliations:** 1Program of Animal Health Technology, Faculty of Veterinary Technology, Kasetsart University, Bangkok 10900, Thailand; authen.pr@ku.th (A.P.); sekbunkorn.tr@ku.th (S.T.); cvtwns@ku.ac.th (W.S.); 2Department of Physiology, Faculty of Veterinary Medicine, Kasetsart University, Bangkok 10900, Thailand; nattaphong.a@ku.th

**Keywords:** antibiotic resistance genes (ARGs), *Arthrospira platensis*, bioremediation, metagenomic sequencing, veterinary hospital wastewater

## Abstract

Veterinary hospital wastewater (VHW) contains high levels of nutrients, organic pollutants, and antibiotic resistance genes (ARGs), posing environmental risks. This study investigated the potential of *Arthrospira platensis* for treating VHW. *A. platensis* was cultivated in different wastewater concentrations for 8 days. Growth and pigment production were highest in 25% VHW, while higher concentrations inhibited performance. In 25% VHW, pigment levels increased significantly compared to the control. Metagenomic analysis showed that cultivation of *A. platensis* in wastewater altered the microbial community and reduced both harmful bacteria and ARGs. The dominant bacteria group, Proteobacteria, decreased markedly after *A. platensis* cultivation in 25% VHW, which was also accompanied by a reduction in key resistance genes. These findings demonstrate that *A. platensis* can simultaneously treat wastewater and produce valuable pigments and biomass, providing a sustainable and environmentally friendly strategy for managing veterinary hospital effluents.

## 1. Introduction

Veterinary hospitals are major sources of antibiotic contamination in wastewater, as broad-spectrum antibiotics are frequently prescribed for companion animals. A wide range of antibiotics has been detected in veterinary hospital wastewater (VHW), with quinolones being the most prevalent group, consistent with their extensive clinical use, followed by β-lactams, tetracyclines, metronidazoles, macrolides, trimethoprim, and sulfonamides, at concentrations ranging from approximately 31 to 15,701 ng/L [1]. This distribution pattern is comparable to reports from human hospital effluents, reflecting the overlap in antibiotic use between human and veterinary medicine [2,3,4]. Despite adherence to antimicrobial stewardship guidelines, drug residues are frequently present in veterinary clinic wastewater, posing environmental risks. These residues may persist through treatment processes and contribute to the development of resistant microorganisms [5,6]. Such bacteria, often carrying transferable resistance genes, can spread through water, waste streams, and areas contaminated by pets. Multidrug-resistant infections place a burden on veterinary care and threaten public health. Addressing this challenge effectively requires the implementation of on-site treatment solutions specifically designed for veterinary wastewater. Conventional wastewater treatment is generally ineffective at removing antibiotics due to their limited treatment capacity [7,8], while alternative methods such as physical separation and chemical degradation are often restricted by high operational costs and economic feasibility [9]. Consequently, there is growing interest in sustainable technologies capable of efficiently removing these contaminants to mitigate environmental and public health risks [10,11,12]. Recent research has focused on developing systems to reduce or eliminate antibiotics prior to discharge or reuse, with increasing attention being directed toward biological alternatives. In particular, microalgae-based systems have emerged as a promising, cost-effective, and environmentally friendly approach for removing various pharmaceuticals and personal care products [13,14].

Microalgal-bacterial consortia (MBC) systems exploit interactions between microalgae and bacteria to simultaneously remove organic matter, excess nutrients, and contaminants [15,16]. These interactions are diverse and complex, ranging from mutualistic symbiosis to competitive antagonism, and may also include commensalism and amensalism [17]. In MBC systems, antagonistic effects can arise when decaying microalgae consume oxygen or release inhibitory compounds, limiting oxygen availability for bacteria. Additionally, under mixotrophic conditions, microalgal consumption of organic matter can further reduce net oxygen production, thereby restricting bacterial activity. Conversely, under a symbiotic relationship, microalgae supply oxygen through photosynthesis to support bacterial nutrient utilization, while bacteria release carbon dioxide that microalgae use. Effective system performance depends on balancing these competing and cooperative dynamics through careful control of light, nutrient levels, and aeration [18,19,20]. *Arthrospira* sp. (commonly known as Spirulina) is particularly promising and has demonstrated strong potential for cultivation in various wastewater sources, including municipal [21], dairy farm [22], and swine wastewater [23]. It can significantly enhance treatment efficiency by utilizing carbon sources in wastewater to support its growth and biomass yield while simultaneously producing valuable bioproducts such as chlorophyll-*a*, phycocyanin, and carotenoids. Although previous studies have demonstrated the benefits of *A. platensis* in wastewater utilization, research on *A. platensis*–bacterial interactions during wastewater treatment, particularly regarding the removal of antibiotic-resistant bacteria and resistance genes, remains limited. Therefore, this study aimed to evaluate the potential of *A. platensis* to produce biomass and pigments in VHW while also assessing its ability to reduce antibiotic-resistant bacteria and resistance genes.

## 2. Materials and Methods

### 2.1. Wastewater Sampling and Quality Analysis

Wastewater samples were randomly collected from a veterinary hospital located at a university in Bangkok, Thailand, in July 2023. Samples were collected in glass bottles with screw caps and transported to the laboratory. Composite sampling was performed by collecting water at hourly intervals over a 3-h period, with each sample taken from the water surface at a depth of approximately 1–2 feet. Sampling points included the center, left edge, and right edge of the wastewater retention pond. Three subsamples of 500 mL each were collected and then combined to create a representative composite sample. The composite sample was filtered through 0.45 μm membrane filters to remove suspended solids and stored at −20 °C until analysis. Physicochemical parameters, including ammonia nitrogen (NH_3_–N), nitrile nitrogen, nitrate nitrogen (NO_3_^−^–N), total phosphate (PO_4_^3−^), biochemical oxygen demand (BOD), chemical oxygen demand (COD), total dissolved solids (TDS), total suspended solids (TSS), total Kjeldahl nitrogen (TKN), fats, oils, and grease (FOG), and pH, were analyzed according to the standard methods for the examination of water and wastewater (APHA/AWWA/WEF) by Central Laboratory (Thailand) Company Limited (Bangkok, Thailand) [24,25].

### 2.2. Microorganism and Culture Conditions

An inoculum was prepared by culturing the *Arthrospira platensis* IFRPD1182, obtained from the Institute of Food Research and Product Development, Kasetsart University, Bangkok, Thailand [26], in Zarrouk medium mixed with non-sterile VHW at 0% VHW (0 mL VHW + 100 mL medium), 25%VHW (25 mL VHW + 75 mL medium), 50%VHW (50 mL VHW + 50 mL medium), 75%VHW (75 mL VHW + 25 mL medium), and 100% (100 mL VHW + 0 mL medium). The Zarrouk medium contained (g/L): NaHCO_3_ 16.8, NaNO_3_ 2.5, K_2_SO_4_ 1.0, NaCl 1.0, K_2_HPO_4_ 0.5, CaCl_2_·2H_2_O 0.04, Na_2_EDTA·2H_2_O 0.03, MgSO_4_·7H_2_O 0.2, FeSO_4_·7H_2_O 0.01, supplemented with 1.0 mL/L of trace elements solution containing (g/L): H_3_BO_3_ 2.86, MoO_3_ 0.02, MnCl_2_·4H_2_O 1.81, CuSO_4_·5H_2_O 0.08, ZnSO_4_·7H_2_O 0.22, Co(NO_3_)_2_·6H_2_O 0.05). The medium was initially adjusted to pH 10 ± 0.5 using 5 N NaOH. Cultures were incubated for 8 days on a rotary shaker at 120 rpm, under continuous LED illumination of 40 µmol/m^2^/s, at 30 ± 2 °C.

### 2.3. Biomass and Pigment Production

Biomass concentration was measured spectrophotometrically (Thermo Fisher Scientific, Waltham, MA, USA), from the optical density of the culture, according to Duangsri [26]. The regression Equation (1) isX (g/L) = 1.2065 × OD_730_ + 0.0079 (R^2^ = 0.998)(1)
where X (g/L) is the biomass concentration and OD_730_ is the absorbance of the suspension at 730 nm, a wavelength chosen to minimize interference from pigments [27].

Chlorophyll-*a* (Chl-*a*) and carotenoids (Car) were quantified using the absorption coefficient (E_λ_) [28], following Equations (2) and (3) [29], respectively. In brief, a 1 mL culture sample was centrifuged at 8000× *g* for 7 min in a high-speed refrigerated microcentrifuge (TOMY, Tokyo, Japan), and the pellet was subsequently extracted with 99.9% methanol in the dark for 20 min at 4 °C. The absorbance of the extracts was measured at 665 nm (λ_max_ for Chl-*a*), 470 nm (λ_max_ for Car), and 720 nm using a Genesys 30 spectrophotometer (Thermo Fisher Scientific, Waltham, MA, USA).Chl-*a* (μg/mL) = 12.9447 × (A_665_ − A_720_)(2)Car (μg/mL) = [1000 × (A_470_ − A_720_) − 2.86 × Chl-*a*]/221(3)

In addition, phycocyanin (PC) and allophycocyanin (APC) were quantified using Equations (4) [30] and (5) [31]. In brief, cells were suspended in phosphate buffer (pH 7), frozen in liquid nitrogen, stored at –20 °C, and disrupted using an ultrasonic homogenizer (Thermo Fisher Scientific, Waltham, MA, USA). The absorbance of extracts was measured at 620 nm (λ_max_ for PC) and 652 nm (λ_max_ for APC) with a Genesys 30 spectrophotometer (Thermo Fisher Scientific, Waltham, MA, USA).PC (μg/mL) = [A_620_ − (0.474 × A_652_)]/5.34(4)APC (μg/mL) = [A_652 −_ (0.208 × A_620_)]/5.09(5)

### 2.4. Shotgun Metagenomic Sequencing and Data Processing

Samples were processed using the ZymoBIOMICS^®^ shotgun metagenomic sequencing service (Zymo Research, Irvine, CA, USA). In brief, wastewater samples were filtered through a 0.2 μm membrane, and DNA was extracted using the ZymoBIOMICS^®^-96 MagBead DNA Kit (Zymo Research Corporation, Irvine, CA, USA). Sequencing libraries were prepared using either the KAPA™ Hyper-Plus Library Preparation Kit (Kapa Biosystems, Wilmington, MA, USA) or the Nextera^®^ DNA Flex Library Prep Kit (Illumina, San Diego, CA, USA), with up to 100 ng of input DNA and internal single-index 8 bp barcodes. Library quantification was performed using TapeStation^®^ (Agilent Technologies, Santa Clara, CA, USA) and quantitative PCR (qPCR), followed by sequencing on a HiSeq^®^ or NovaSeq^®^ platform (Illumina, San Diego, CA, USA). Raw sequencing reads were quality-trimmed using Trimmomatic-0.33 [32]. Taxonomic identification of bacterial and archaeal sequences was conducted using the complete GTDB database (R07-RS207) and pre-formatted GenBank databases (v.2022.03) via Sourmash.

### 2.5. Statistical Analysis

All data obtained in this study represent the means of three independent biological replicates, and the error bars represent the standard deviation (Mean ± SD, *n* = 3). The statistical analysis was analyzed by one way analysis of variance (ANOVA) and the significant difference (*p* < 0.05) were compared by Tukey’s HSD using SPSS version 22 (IBM, New York, NY, USA).

## 3. Results

### 3.1. Water Quality of VHW

Wastewater samples collected from the veterinary hospital were analyzed for key physicochemical parameters, as summarized in Table 1. The results indicated considerable contamination, particularly with nitrogenous compounds, organic pollutants, and solids.

### 3.2. Effect of VHW Proportions on Biomass and Pigment Production of A. platensis

As shown in Figure 1a, *A. platensis* cultured in 25% and 50% VHW exhibited growth patterns similar to 0% VHW but with notably higher OD_730_ values over the 8-day cultivation period. On day 8, biomass concentrations were 0.58 ± 0.04 g/L (0% VHW, control), 0.78 ± 0.05 g/L (25% VHW), 0.65 ± 0.05 g/L (50% VHW), and 0.50 ± 0.05 g/L (75% VHW). Biomass in the 25% VHW treatment was significantly higher than in the control (0% VHW) (*p* < 0.05), while the 50% and 75% VHW treatments did not differ significantly from the control. By contrast, the 100% VHW condition strongly inhibited growth, with marked suppression observed during the first two days and complete cessation by day 4 (Figure 1b).

Figure 2a–d show pigment production levels of chlorophyll-*a* (Chl-*a*), carotenoids (Car), phycocyanin (PC), and allophycocyanin (APC) in *A. platensis* cultures. All pigment levels in cultures with 25% VHW peaked on day 8, showing significant increases (*p* < 0.05) compared to 0% VHW: 1.3-fold for Chl-*a* (12.0 μg/mL), 1.5-fold for Car (4.4 μg/mL), 1.7-fold for PC (120 μg/mL), and 2.3-fold for APC (54 μg/mL). The 50% VHW also led to higher pigment levels than the control, though lower than those observed at 25% VHW. By contrast, cultures grown in 75% and 100% VHW exhibited reduced pigment content, with 100% VHW showing negligible pigment production.

### 3.3. Metagenomic Analysis of Wastewater Samples from a Veterinary Hospital

Shotgun metagenomic sequencing was employed to investigate microbial communities in VHW. Three sample types were examined: 100% VHW (raw VHW), 25% VHW cultured without *A. platensis* (T25), and 25% VHW cultured with *A. platensis* (T25AP). Taxonomic composition at the phylum, order, and genus levels was analyzed for both eukaryotic and prokaryotic communities. In raw VHW, the dominant eukaryotic phyla (≥10% relative abundance) were Basidiomycota (12.8%) and Apicomplexa (27.6%) (Figure 3a). Phyla with lower relative abundance included Evosea (9.8%), Oomycota (9.7%), Ascomycota (5.7%), and Ciliophora (6.4%). Under the T25 condition, the dominant eukaryotic phyla were Evosea (26.7%), Basidiomycota (16.8%), Oomycota (15.4%), and Apicomplexa (15.3%). By contrast, under the T25AP condition, Oomycota was no longer present, while Evosea (42.0%), Basidiomycota (44.0%), and Apicomplexa (14.0%) increased in relative abundance.

At the order level, dominant eukaryotic taxa (≥10% relative abundance) in raw VHW were *Eucoccidiorida* (27.6%) and *Chromulinales* (26.7%) (Figure 3a). Orders with lower relative abundance were Pucciniales (6.4%), Periculida (6.4%), Acytosteliales (9.8%), Peronosporales (9.7%), Erysiphales (4.8%), Boletales (2.7%), and Polyporales (3.3%). Under the T25 condition, dominant orders included Acytosteliales (26.7%), Chromulinales (19.9%), Pucciniales (16.8%), Peronosporales (15.4%), and Eucoccidiorida (15.3%), while Boletales and Polyporales were no longer detected. Following T25AP cultivation, Peronosporales and Chromulinales disappeared, while Acytosteliales (42.0%), Pucciniales (44.0%), and Eucoccidiorida (14.0%) increased in relative abundance.

At the genus level, dominant eukaryotic genera (≥10% relative abundance) in raw VHW included *Cyclospora* sp. (24.8%) and *Chromulinospumella* sp. (17.7%). Genera with lower abundance were *Acytostelium* sp. (9.8%), *Pseudoperonospora* sp. (9.7%), *Puccinia* sp. (6.4%), and *Paramecium* sp. (6.4%) (Figure 3a). Under the T25 condition, the dominant genera shifted to *Dinobryon* sp. (19.9%), *Puccinia* sp. (16.8%), *Pseudoperonospora* sp. (15.4%), *Cyclospora* sp. (15.3%), *Rostrostelium* sp. (14.4%), and *Acytostelium* sp. (12.4%). By contrast, under the T25AP condition, *Dinobryon* sp. and *Pseudoperonospora* sp. disappeared, while *Cyclospora* sp. (14.0%), *Puccinia* sp. (44.0%), and *Acytostelium* sp. (42.0%) showed pronounced increases in relative abundance.

In addition, in raw VHW, the dominant prokaryotic phyla (≥10% relative abundance) were Bacteroidota (31.5%), Proteobacteria (25.4%), Campylobacterota (11.4%), and Firmicutes_A (11.0%). Phyla with lower relative abundance included Desulfobacterota_I (8.5%), Desulfobacterota (3.5%), and Actinobacteriota (2.4%) (Figure 3b). Under the T25 condition, the dominant prokaryotic phylum was Proteobacteria (97.0%). By contrast, under the T25AP condition, Proteobacteria (11.6%), Bacteroidota (2.9%), and Firmicutes_A (0.1%) decreased markedly in relative abundance.

At the order level, dominant prokaryotic taxa (≥10% relative abundance) in raw VHW were Bacteroidales (31.4%), Burkhoderiales (19.2%), and Campylobacterales (11.4%) (Figure 3b). Orders with lower relative abundance were Desulfovibrionales (8.5%), Lachnospirales (4.8%), Eneterobacterales (3.6%), Oscillospirales (3.3%), and Pseudomonadales (2.6%). Under the T25 condition, dominant orders included Pseudomonadales (27.9%), Nitrosococcales (23.5%), Caulobacterales (17.8%), and Rhodobacterales (10.4%), while Rhizobiales (9.3%) and Sphingomonadales (4.3%) were detected at lower relative abundance. Following T25AP cultivation, all these orders decreased in relative abundance.

At the genus level, dominant prokaryotic genera (≥10% relative abundance) in raw VHW included *Aliarcobacter* sp. (10.4%). Genera with lower abundance were *Bacteroides* sp. (6.0%), *Phocaeicola* sp. (6.0%), *Prevotella* sp. (5.7%), *Thauera* sp. (8.1%), *Azonexus* sp. (5.1%), and *Desulfomicrobium* sp. (7.9%) (Figure 3b). Under the T25 condition, the dominant shifted to *Methylophaga* sp. (23.5%), *Pseudomonas_A* sp. (16.8%), *Oceanicaulis* sp. (17.7%), and *Roseinatronobacter* sp. (10.4%), while *Thauera* sp., *Bacteroides* sp., *Desulfomicrobium* sp., *Azonexus* sp., *Phocaeicola* sp., and *Prevotella* sp. disappeared. Following T25AP cultivation, *Methylophaga* sp. and *Pseudomonas_A* sp. were no longer present, while *Oceanicaulis* sp. (4.7%) and *Roseinatronobacter* sp. (1.5%) decreased in relative abundance.

### 3.4. Distribution of Antibiotic Resistance Genes (ARGs) Across Bacterial Genera

The heatmap in Figure 4 illustrates the distribution of ARGs across bacterial genera under two cultivation conditions: 25% VHW without *A. platensis* (T25) and 25% VHW with *A. platensis* (T25AP). The *x*-axis represents bacterial genera, each with columns for T25 and T25AP, while the *y*-axis lists 20 ARGs, including *aph*, *oqxB*, *ant*, *aac*, *msr*, *mph*, *qac*, *vat*, *tet*, *erm*, *mef*, *dfr*, *sul*, *cat*, *lnu*, and several beta-lactamase genes such as *bla*_OXA_, *bla*_PBR_, and *bla*_NPS-1_. Notably, *Acinetobacter* sp. under the T25 condition exhibited high expression of *aph* and moderate levels of *oqxB* and *bla_OXA_*, while these genes were less pronounced under T25AP. *Methylophaga* sp. displayed the strongest signal for *oqxB* under T25, suggesting a high prevalence of resistance genes. *Pseudomonas_E* sp. showed elevated *oqxB* under T25 and some *mph* expression under T25AP. Other genera, such as *Halomonas* sp., *Phoceicola* sp., and *Thauera* sp., demonstrated moderate levels of *mph*, *erm*, and *tet*, with slightly increased abundance under T25AP in some cases. By contrast, *Bifidobacterium* sp., *Blautia*_A sp., and *Gemmiger* sp. showed low or minimal ARG presence, indicating a limited role in resistance dissemination. Interestingly, *Tolumonas* sp. exhibited moderate *aph* abundance under both conditions. Overall, the heatmap reveals genus-specific ARG distribution patterns and highlights shifts in gene abundance under *A. platensis* treatment.

## 4. Discussion

This study investigated the potential of *A. platensis* as an alternative approach for removing ARGs while simultaneously producing valuable pigments from VHW. Antibiotics pose significant risks to both environmental and human health, even at low concentrations, and their overuse and misuse have contributed to the emergence of antibiotic-resistant bacteria, reducing the effectiveness of commonly prescribed antimicrobial therapies. Previous reports have demonstrated the occurrence of various classes of antibiotics in VHW, including quinolones, penicillin, tetracyclines, metronidazoles, macrolides, trimethoprim, and sulfonamides [1,7]. In the present study, however, antibiotic concentrations in VHW were not directly measured. Nevertheless, the antibiotics present are likely to include common veterinary drugs given their widespread therapeutic use in animals. The physicochemical profile of the VHW provides essential context for understanding the observed effects on *A. platensis* growth and pigment production (Table 1). Elevated levels of NH_3_–N (56.56 ± 3.23 mg/L) and TKN (76.13 ± 3.10 mg/L) indicate a nitrogen-rich environment, favorable for *A. platensis* cultivation, since nitrogen is required for protein synthesis and the biosynthesis of pigments such as phycobiliproteins and chlorophylls [33,34]. The low concentration of nitrate and undetectable nitrite suggest limited nitrification, possibly resulting from the anaerobic nature of the wastewater or the absence of active nitrifying bacteria at the time of sampling. At 25% VHW, the nitrogen load appears optimal, supporting biomass accumulation and pigment synthesis. This dilution likely provides sufficient nitrogen while minimizing toxicity. Conversely, inhibition observed at 75% and 100% VHW is likely due to excessive free ammonia, which at high concentrations disrupts cellular pH homeostasis and enzyme activity, impairing metabolic functions [35].

The high concentrations of organic pollutants, reflected by elevated BOD (71.50 ± 12.31 mg/L) and COD (160.97 ± 24.44 mg/L), indicate a considerable organic load that may indirectly affect microalgal growth. While *A. platensis* is not primarily heterotrophic, the presence of organic matter may promote the growth of competing heterotrophic bacteria in non-sterile cultures, resulting in nutrient competition or production of inhibitory metabolites. At high wastewater concentrations, such microbial competition and oxygen depletion could impair *A. platensis* viability. Additionally, the PO_4_^3−^ concentration (6.31 ± 1.43 mg/L) supports pigment biosynthesis and growth under light-sufficient conditions, as phosphorus is vital for ATP production and nucleic acid synthesis [36]. Excessive phosphorus in undiluted wastewater, however, may disrupt nutrient ratios (e.g., N:P), exacerbating stress responses [37]. The presence of FOG (9.59 ± 2.58 mg/L), together with moderate TDS (412.00 ± 21.57 mg/L) and TSS (25.06 ± 1.83 mg/L), may further explain the inhibition observed under concentrated VHW conditions. Lipophilic substances may coat microalgal cells, hindering nutrient uptake and light absorption, while suspended solids reduce light penetration, promoting shading effects in the culture medium.

This study demonstrates that diluted VHW can serve as a viable supplement to conventional media for cultivating *A. platensis*, enhancing both biomass and pigment production at optimal concentrations [38]. The 25% VHW treatment produced the highest biomass (0.78 ± 0.05 g/L) (Figure 1). These results suggest that moderate dilution provides adequate nitrogen and phosphorus for growth while minimizing toxicity from raw wastewater. Biomass at 50% VHW (0.65 ± 0.05 g/L) was slightly lower but still exceeded the control, indicating the adaptability of *A. platensis* to moderate wastewater concentrations. By contrast, growth in 100% VHW ceased by day 4, likely due to high organic loads or residual antibiotics impairing photosynthesis, enzymatic activity, and metabolism [37]. These findings are consistent with earlier studies reporting that high wastewater concentrations induce oxidative or chemical stress in microalgae, inhibiting growth [39].

Pigment production by *A. platensis* in response to varying VHW concentrations highlights its ability to utilize nutrients in diluted wastewater for metabolism and biosynthesis. At 25% VHW, pigment levels of chlorophyll-*a*, carotenoids, phycocyanin, and allophycocyanin increased significantly (Figure 2), indicating that moderate dilution supplies nutrients without triggering stress. On day 8, pigment levels peaked, with phycocyanin and allophycocyanin showing 1.7- and 2.3-fold increases, respectively, over the control, likely due to sufficient nitrogen and phosphorus availability. These results align with previous studies demonstrating enhanced pigment synthesis under low to moderate nutrient enrichment [40]. Higher VHW concentrations (75% and 100%) suppressed pigment production, with 100% VHW yielding negligible levels, likely due to toxic effects from residual pollutants [41]. Overall, the results emphasize the importance of partial wastewater dilution in facilitating nutrient uptake by *A. platensis*.

Shotgun metagenomic analysis revealed significant shifts in microbial community composition in VHW [42]. In raw VHW, the dominant eukaryotic phyla (≥10% relative abundance) were Basidiomycota (12.8%) and Apicomplexa (27.6%), while Bacteroidota (31.5%) and Proteobacteria (25.4%) dominated among prokaryotes (Figure 3). Under T25 conditions (25% VHW without *A. platensis*), eukaryotic phyla such as Evosea (26.7%) and Oomycota (15.4%) increased, while Basidiomycota (16.8%) and Apicomplexa (15.3%) decreased slightly. By contrast, the prokaryotic community shifted strongly toward Proteobacteria (97.0%), while Bacteroidota declined sharply (0.9%). Genus-level changes showed increases in *Dinobryon* sp., *Puccinia* sp., *Pseudoperonospora* sp., *Cyclospora* sp., *Rostrostelium* sp., and *Acytostelium* sp., while dominant prokaryotes included *Methylophaga* sp., *Pseudomonas_A* sp., *Oceanicaulis* sp., and *Roseinatronobacter* sp. Moreover, several moderate genera disappeared. This shift likely resulted from the alkaline pH (~10) and high salinity of the Zarrouk-based medium, which imposed osmotic stress and restricted bacterial growth. This medium relies on inorganic carbon sources, such as NaHCO_3_, and lacks the organic carbon required by many bacterial species, thereby suppressing sensitive groups. Exceptions include alkaliphilic or halotolerant bacteria, which may still grow under these conditions [43]. In addition, co-cultivation with *A. platensis* (T25AP) further reshaped the community, with Oomycota disappearing while Evosea (42.0%) and Basidiomycota (44.0%) increased in abundance. Among prokaryotes, Proteobacteria decreased to 11.6% and Bacteroidota to 2.9%. Certain eukaryotic genera (*Dinobryon* sp., *Pseudoperonospora* sp.) and prokaryotic genera (*Methylophaga* sp., *Pseudomonas_A* sp.) disappeared, while *Cyclospora* sp., *Puccinia* sp., and *Acytostelium* sp. markedly increased among eukaryotes. Similarly, *Oceanicaulis* sp. and *Roseinatronobacter* sp. decreased in abundance among prokaryotes. These results indicate that *A. platensis* exerts strong selective pressure on both eukaryotic and prokaryotic communities, likely through nutrient competition, enhanced oxygenation, and production of inhibitory compounds [44], highlighting its potential for suppressing undesirable or antibiotic-resistant microorganisms in wastewater treatment.

The distribution of ARGs across bacterial genera under T25 and T25AP conditions showed notable shifts in resistance profiles that may be attributed to antibiotic pressure. The presence and abundance of specific ARGs varied significantly depending on both the bacterial genus and treatment condition, suggesting a complex interplay between microbial composition, antibiotic exposure, and gene mobility [45,46,47]. One of the most striking observations was the high abundance of the *aph* gene in *Acinetobacter* sp. under the T25 condition, which decreased under T25AP. This pattern may reflect either a selective reduction in *Acinetobacter* sp. populations carrying the *aph* gene or suppression of expression due to competition with other microbial taxa under antibiotic pressure. The *aph* gene is known to confer resistance to aminoglycosides, and its strong signal under baseline conditions highlights the potential of *Acinetobacter* sp. as a reservoir of aminoglycoside resistance genes [48]. The concurrent detection of *oqxB* and *bla*_OXA_ in *Acinetobacter* sp. further emphasizes its multidrug resistance potential. *Methylophaga* sp. exhibited the highest *oqxB* abundance under T25, which sharply declined under T25AP. The *oqxB* gene, often associated with efflux pumps conferring resistance to quinolones and other drugs, may be downregulated or lost due to competitive exclusion or horizontal gene transfer dynamics under antibiotic stress [49]. Interestingly, *Pseudomonas_E* sp. showed a similar pattern, with elevated *oqxB* levels in T25, supporting the idea that this gene is prevalent among Gram-negative opportunistic pathogens under non-stressed conditions. The T25AP treatment led to increased detection of several resistance genes across multiple genera, indicating a possible enrichment of ARG-harboring populations in response to antibiotic pressure. For instance, *mph*, *erm*, and *tet* genes, which typically confer resistance to macrolides, lincosamides, and tetracyclines [50], were more widely distributed or showed higher abundance in genera such as *Thauera* sp., *Phoceicola* sp., and *Bifidobacterium* sp. under T25AP. This suggests that these taxa may harbor inducible resistance mechanisms or have acquired ARGs through horizontal gene transfer when exposed to antibiotics.

The increased abundance of *tet* genes in *Comamonas* sp., *Proteiniclasticum* sp., and *Phoceicola* sp. under both T25 and T25AP conditions indicates that these genera may naturally harbor or maintain stable tetracycline resistance genes regardless of antibiotic exposure. On the other hand, some genera such as *Gemmiger* sp. and *Blautia_A* sp. showed minimal ARG presence, aligning with their classification as primarily gut-associated or anaerobic bacteria with limited antibiotic resistance potential [51]. Interestingly, *Tolumonas* sp. consistently expressed the *aph* gene under both treatments, possibly indicating stable aminoglycoside resistance or a niche that enables ARG maintenance even in the absence of selective pressure. This persistent presence may suggest intrinsic resistance or a conserved genetic element. It is also noteworthy that beta-lactam resistance genes, including *bla*_OXA_, *bla*_LRC-1_, and *bla*_NPS-1_, were detected at low levels. This may reflect the specificity of antibiotic selection pressure in T25AP, which might not have directly favored beta-lactam resistance, or the limited dissemination of these genes among the dominant bacterial populations in this experimental setup. Altogether, the data underscore how antibiotic exposure can reshape microbial community structure and enhance the dissemination or expression of specific ARGs. The variation in gene profiles between T25 and T25AP across bacterial genera emphasizes the need for targeted analyses to better understand resistance dynamics in microbial ecosystems.

These results indicate that *A. platensis* can play a key role in mitigating the spread of ARGs by altering environmental conditions and shaping microbial community composition. Although ARGs remained detectable, their abundance was reduced under *A. platensis* treatment, highlighting its potential in wastewater treatment. Future work should explore system scale-up, the mechanisms of antibiotic and ARG removal, and potential applications for the treated microalgal biomass.

## 5. Conclusions

This study demonstrates the potential of *Arthrospira platensis* for treating veterinary hospital wastewater. Cultivation in diluted wastewater not only enhanced biomass and pigment production but also significantly altered microbial communities and reduced antibiotic resistance gene abundance. These findings highlight *A. platensis* as a promising candidate for integrated wastewater treatment, offering both environmental and biotechnological benefits.

## Figures and Tables

**Figure 1 biology-14-01396-f001:**
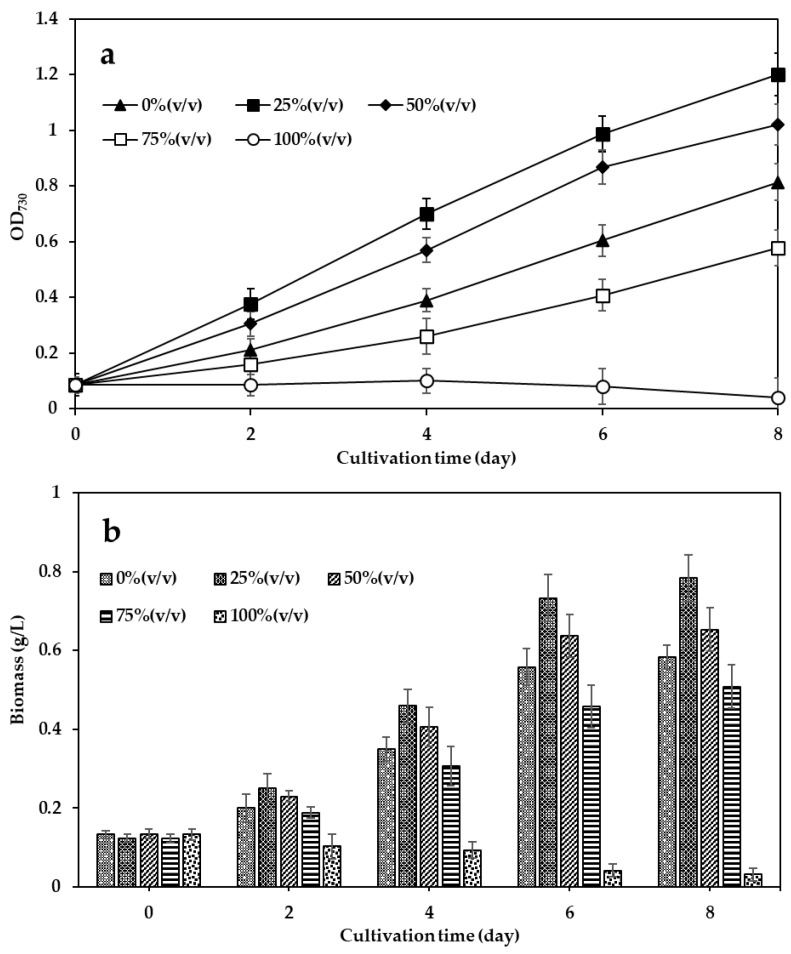
Growth of *A. platensis* cultivated in Zarrouk medium mixed with different proportions of VHW over an 8-day period. (**a**) Growth curves based on optical density at 730 nm (OD_730_) for cultures grown in 0%, 25%, 50%, 75%, and 100% VHW, and (**b**) Biomass concentrations (g/L) under each treatment condition. Data are expressed as Mean ± SD (*n* = 3).

**Figure 2 biology-14-01396-f002:**
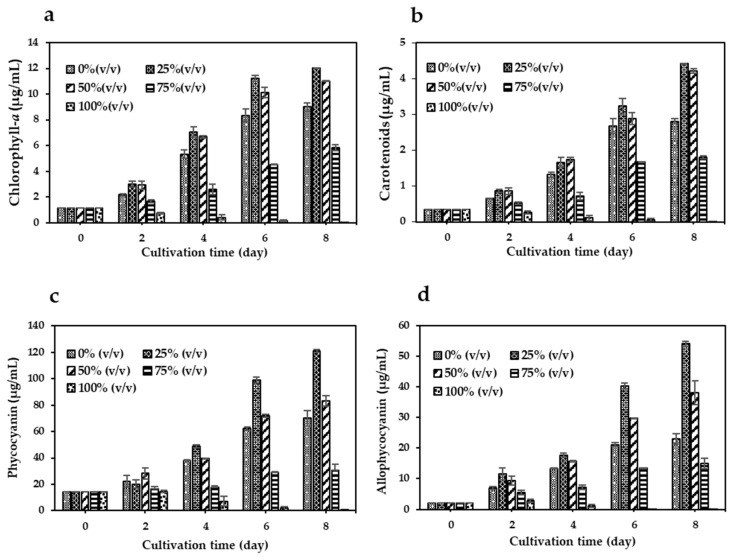
Pigment production in *A. platensis* cultivated in Zarrouk medium mixed with different proportions of VHW over an 8-day period. (**a**) Chlorophyll-*a* (Chl-*a*), (**b**) carotenoids (Car), (**c**) phycocyanin (PC), and (**d**) allophycocyanin (APC) concentrations in 0%, 25%, 50%, 75%, and 100% VHW. Data are expressed as Mean ± SD (*n* = 3).

**Figure 3 biology-14-01396-f003:**
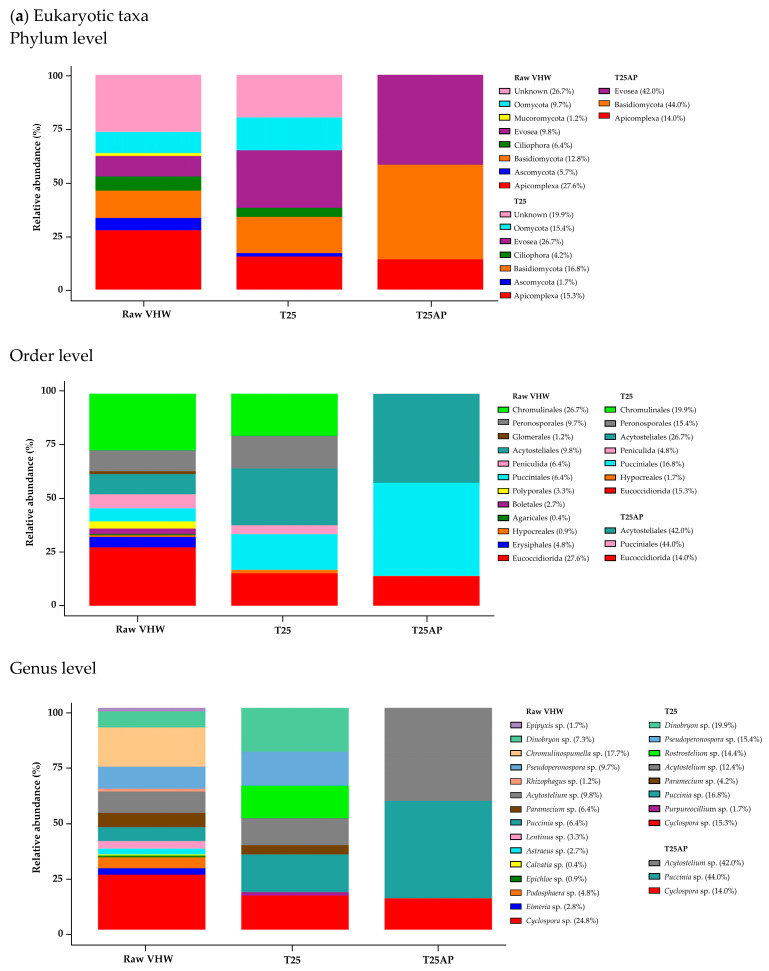
Microbial community composition in raw VHW, 25% VHW without *A. platensis* (T25), and 25% VHW with *A. platensis* (T25AP), as determined by shotgun metagenomic sequencing. (**a**) Relative abundance of dominant eukaryotic taxa at the phylum, order, and genus levels; (**b**) relative abundance of dominant prokaryotic taxa at the phylum, order, and genus levels.

**Figure 4 biology-14-01396-f004:**
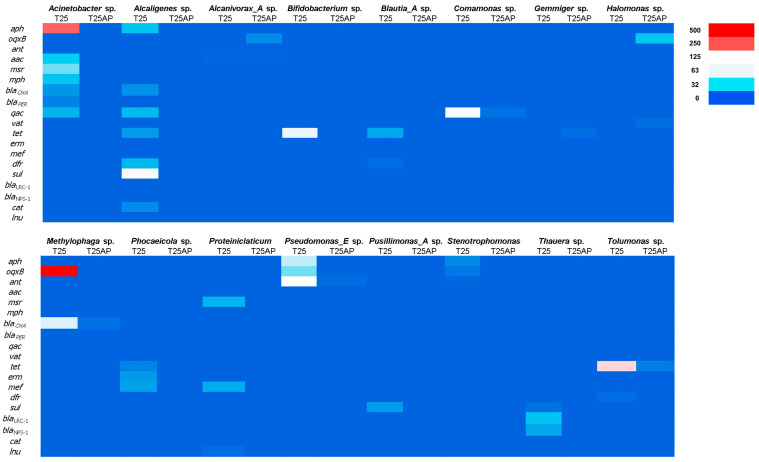
Distribution of antibiotic resistance genes (ARGs) across bacterial genera in 25% VHW without *A. platensis* (T25) and 25% VHW with *A. platensis* (T25AP), based on shotgun metagenomic analysis. The color intensity, ranging from deep blue (low abundance) to red (high abundance), reflects the relative abundance of each ARG.

**Table 1 biology-14-01396-t001:** Characterization of the sampled VHW.

Parameters	Values
NH_3_–N	56.56 ± 3.23 mg/L
NO_3_^−^–N	≤0.011 mg/L
PO_4_^3−^	6.31 ± 1.43 mg/L
BOD	71.50 ± 12.31 mg/L
COD	160.97 ± 24.44 mg/L
TDS	412.00 ± 21.57 mg/L
TSS	25.06 ± 1.83 mg/L
TKN	76.13 ± 3.10 mg/L
FOG	9.59 ± 2.58 mg/L
pH	7.5 ± 0.2

## Data Availability

The data supporting the findings of this study are available from the corresponding author upon reasonable request.

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
