# Peer review of "Cultivation of Arthrospira platensis in Veterinary Hospital Wastewater Enhances Pigment Production and Reduces Antibiotic Resistance Genes"

_biology, 2025, doi:10.3390/biology14101396_

Round 1

Reviewer 1 Report

Comments and Suggestions for Authors

The present study evaluated the bioremediation potential of Arthrospira platensis in treating veterinary hospital wastewater (VHW), which contains high levels of nutrients, pollutants, and antibiotic resistance genes (ARGs). Following comments may be incorporated to improve the manuscript.

  1. Lines 2–4: The title should be modified to improve clarity and accurately reflect the study's scope.

  2. Line 29: Use standard abbreviations for Biological Oxygen Demand (BOD) and Chemical Oxygen Demand (COD).

  3. Lines 37 and 39: Avoid using unnecessary sample names like T25 and T25AP in abtract.

  4. Lines 50–52: Provide the concentration range of antibiotics detected in VHW, and support this with appropriate citations.

  5. Line 97: Antibiotic concentrations should have been quantified to better evaluate their effect on Arthrospira platensis.

  6. Line 103: Replace “millileter” with the correct unit “mL”.

  7. Line 113: Specify the source and supplier of Arthrospira platensis strain IFRPD1182.

  8. Line 114: Use the abbreviation “VHW” consistently throughout the manuscript after its initial definition.

  9. Line 115: Since Zarrouk medium lacks a carbon source, control culture growth may be suppressed. Clarify how meaningful comparisons were made between control and VHW-fed cultures.

  10. Lines 117–118: Include missing units for all trace elements (e.g., H₃BO₃, MnCl₂·4H₂O, etc.).

  11. Line 118: Provide details of the equipment used for incubation (e.g., type of shaker, incubator).

  12. Line 161: Table 1 lacks pH values. Include pH data to complete the physicochemical characterization.

  13. Lines 164–167: The sentence describing cultivation and periodic measurements repeats methodological information already provided and should be removed to avoid redundancy.

  14. Lines 257–264: These lines also repeat information from the methodology and should be deleted to maintain clarity and avoid repetition.

Author Response

Response to Reviewers

We would like to express our sincere gratitude to the reviewers for their thorough evaluation and constructive feedback on our manuscript. Their insightful comments and suggestions have been invaluable in improving the overall quality, clarity, and scientific rigor of the revised version. In the following section, we provide detailed, point-by-point responses to all comments, and we trust that our revisions adequately address the reviewers’ concerns.

Reviewer I’ s comments

 The present study evaluated the bioremediation potential of Arthrospira platensis in treating veterinary hospital wastewater (VHW), which contains high levels of nutrients, pollutants, and antibiotic resistance genes (ARGs). Following comments may be incorporated to improve the manuscript.

 1. Lines 2–4: The title should be modified to improve clarity and accurately reflect the study's scope.

Authors response: We have revised the title to make it more consistent with the scope of the research. The title has been changed to Cultivation of Arthrospira platensis in Veterinary Hospital Wastewater Enhances Pigment Production and Reduces Antibiotic Resistance Genes (Lines 2-4).

2.  Line 29: Use standard abbreviations for Biological Oxygen Demand (BOD) and Chemical Oxygen Demand (COD).

Authors response: We have revised the terms by changing BOD and COD to biological oxygen demand (BOD) and chemical oxygen demand (COD), as recommended (Lines 30-31).

3. Lines 37 and 39: Avoid using unnecessary sample names like T25 and T25AP in abstract.

Authors response: We have revised the sentence and removed the abbreviation, T25 and T25AP in abstract.

4. Lines 50–52: Provide the concentration range of antibiotics detected in VHW, and support this with appropriate citations.

 Authors response: We have incorporated the concentration range of antibiotics detected in VHW, as reported in a previous study, and have provided the appropriate citation to support this revision (Lines 50–54).

5. Line 103: Replace “milliliter” with the correct unit “mL”.

 Authors response: As suggested, we have corrected “milliliter” to the appropriate abbreviation “mL” (Line 105).

6. Line 97: Antibiotic concentrations should have been quantified to better evaluate their effect on Arthrospira platensis.

Authors response: In the present study, we did not perform quantification of antibiotic concentrations.

7. Line 113: Specify the source and supplier of Arthrospira platensisstrain IFRPD1182.

 Authors response: We have included information on the source and supplier of Arthrospira platensis IFRPD1182, obtained from the Institute of Food Research and Product Development, Kasetsart University, Bangkok, Thailand (Lines 116-118).

8. Line 114: Use the abbreviation “VHW” consistently throughout the manuscript after its initial definition.

 Authors response: We have corrected and ensured the consistent use of the abbreviation “VHW” throughout the manuscript.

9. Line 115: Since Zarrouk medium lacks a carbon source, control culture growth may be suppressed. Clarify how meaningful comparisons were made between control and VHW-fed cultures.

 Authors response: The control culture was grown in Zarrouk medium, which is specifically formulated for Arthrospira (Spirulina). This medium contains inorganic carbon sources such as NaHCO₃ but lacks the organic carbon required by many bacteria, thereby allowing only alkaliphilic and certain other bacteria to grow selectively. In contrast, Zarrouk medium mixed with non-sterile VHW contained NaHCO₃ supplemented with varying proportions of VHW, which provides organic pollutants and nutrients that may support broader microbial growth. Comparisons between the control and VHW-mixed cultures are therefore meaningful, as any increase in growth in the VHW treatments indicates the presence of additional nutrients in the wastewater that can support both bacterial growth and A. platensis cultivation. (Zarrouk medium mixed with non-sterile VHW at 0% VHW (0 mL VHW + 100 mL medium), 25%VHW (25 mL VHW + 75 mL medium), 50%VHW (50 mL VHW + 50 mL medium), 75%VHW (75 mL VHW + 25 mL medium), and 100% (100 mL VHW + 0 mL medium).

10. Lines 117–118: Include missing units for all trace elements (e.g., H₃BO₃, MnCl₂4H₂O, etc.).

 Authors response: The sentence has been revised to include the unit g/L (Lines 124), as suggested.

11. Line 118: Provide details of the equipment used for incubation (e.g., type of shaker, incubator).

 Authors response: We have provided details of the instruments used (Lines 126-127), as suggested. Cultures were incubated for 8 days on a rotary shaker at 120 rpm, under continuous LED illumination of 40 µmol/m²/s, at 30 ± 2 °C.

12. Line 161: Table 1 lacks pH values. Include pH data to complete the physicochemical characterization.

 Authors response: We have added the pH value (7.5 ± 0.2) to Table 1.

13. Lines 164–167: The sentence describing cultivation and periodic measurements repeats methodological information already provided and should be removed to avoid redundancy.

 Authors response: We have removed the sentence to avoid redundancy, as suggested.

14. Lines 257–264: These lines also repeat information from the methodology and should be deleted to maintain clarity and avoid repetition.

 Authors response: We have deleted the sentence to prevent repetition of methodological information, as suggested.

Reviewer 2 Report

Comments and Suggestions for Authors

General comment

The abbreviation VHW is introduced to the text several time (Lines 25, 165, 187, 193, 199, 227, 263). The situation should be avoided. The authors should use this abbreviation in the whole text (please, check the Lines 114, 149, 161, 162, 396, 407).

Abstract

Lines 30-31: The phrase “A. platensis was cultivated for 8 days in Zarrouk medium supplemented with 0–100% VHW.” is not clear.  Same situation is in Lines 114-115: “Zarrouk medium diluted with veterinary hospital wastewater at 0%, 25%, 50%, 114 75%, and 100% concentrations (v/v). “ Was there replacement of Zarrouk medium by the VHW per 0-100%? Replacement but not dilution? I suggest that the authors talk about diluting VHW using Zarrouk medium, since in many places in the text the authors use 25%VHW".

Lines 39-40 (Abstract) and Lines 83-84 (in Introduction): Text of abstract (actual results obtained) on significant suppression of Proteobacteria cell growth and the motivation presented in the Introduction, according to which the biological role of microalgal-bacterial symbiosis is crucial for the establishment and stability of the symbiotic system" contradict each other. There is a negative symbiosis for bacteria which was revealed by the authors in this work. In this regard, I recommend that the authors redo the text in the Introduction (Lines 75-87). It is necessary to focus on those studies that have already shown in other works that in addition to the possibility of accumulation of microalgae biomass containing pigments and reduction of nitrogen- and carbon-containing pollutants in wastewater, there is a significant antimicrobial effect of microalgae on bacterial cells present in wastewater.

Methods

Lines 106-111: Please, add the information about equipment used to realize all analytical methods in this work for the determination of ammonia nitrogen (NH₃-N), nitrile nitrogen, nitrate nitrogen (NO₃⁻-N), total phosphate (PO₄³⁻), biochemical oxygen demand (BOD), chemical oxygen demand (COD), total dissolved solids (TDS), total suspended solids (TSS), total Kjeldahl nitrogen (TKN), fats, oils, grease (FOG) and pH.

Line 119-120: Please, add the equipment (type of photobioreactor) used for cell cultivation, intensity and regime (day/night or constant) of illumination.

Line 126: Please add the type of centrifuge.

Lines 128: Please add the equipment used for measurements of absorbance at 665, 470, and 720 nm.

Please, the information about the methods of statistical treatment of obtained results in this work. It is important because there are following results in the text (Lines 171-173): “ 0.65 ± 0.05 g/L in the 50% VHW condition, and 0.58 ± 0.04 g/L) in control. Biomass production declined under the 75% VHW condition, resulting in a final concentration of 0.50 ± 0.05 g/L.” According to ANOVA there is no significant difference between results obtained for 50% VHW and Control, Control and 75% VHW…Please, see.

Results

Lines 152-159: The following text “Ammonia nitrogen and total Kjeldahl nitrogen (TKN) were found at concentrations of 56.56 ± 3.23 mg/L and 76.13 ± 3.10 mg/L, respectively, while nitrate nitrogen levels were very low (≤ 0.011 mg/L). Nitrile nitrogen was not detected. The samples also contained notable levels of organic pollutants, with a biochemical oxygen demand (BOD) of 71.50 ± 12.31 mg/L and a chemical oxygen demand (COD) of 160.97 ± 24.44 mg/L. Total phosphate was measured at 6.31 ± 1.43 mg/L. Regarding solids content, total dissolved solids (TDS) and total suspended solids (TSS) were recorded at 412.00 ± 21.57 mg/L and 25.06 ± 1.83 mg/L, respectively. Additionally, fat, oil, and grease were present at a concentration of 9.59 ± 2.58 mg/L.” should be considerable redone, because it contains same information which is performed in the Table 1. So, there is obvious doubling of same information in the article. The numbers shown in Table 1 should be deleted from the text.

Lines164-167: The textA. platensis was cultivated in 50 mL Zarrouk medium and in Zarrouk medium sup-164 plemented with non-sterile veterinary hospital wastewater (VHW) at varying ratios: 25%, 165 50%, 75%, and 100% (v/v). Cultures were maintained under continuous illumination 166 (40 μmol/m²/s) for 8 days, with biomass and pigment contents recorded every two days.” repeats the text performed in section 2.2 (Methods), and here the authors mentioned cultivation for 8 days, whereas 7 days they gave in Line 118 (contradiction!).

Figure 1(a) and figure capture: Please, check “Growth curves based on optical density at 730 nm (OD730)”. Are the authors sure that they controlled the growth of cells but not the chlorophyll-a accumulation at such wave length? Please, be correct and give the reference with recommendation of use such OD730 to control the cell biomass but not a fluorescent pigment. Please, see the Lines 128 and 130: there is the text with mentioning of 720 nm, but not 730 nm, and the extracts (!) from cells are discussed but not cell suspensions for estimation of biomass.

Figure 2. Please, see that the concentration of Chlorophyll-a (µg/mL) is 1,000 times less as compared to carotenoids (mg/mL)! It is impossible, because A. platensis cells have green color, but not red which is typical for carotenoids. Some unreal results are performed here. To check the situation, please see any other paper with the same cells. For example, this: DOI: 10.1039/C9GC03292D (Carotenoids, chlorophylls and phycocyanin from Spirulina: supercritical CO2 and water extraction methods for added value products cascade. Green Chem., 2020, 22, 187-196.).

Figure 3. There is NO very important control: the results of cultivation of 25%VHW with microorganisms initially present in this sample for 8 days without A. platensis! The pH of the medium used for cultivation was 10.0 ± 0.5 (see Line 119). This pH value is not convenient for the bacterial cells, so the action of the medium but not the presence of A. platensis for 8 days without fresh substrates for the bacteria could result in the decrease of their amounts. In fact, the antimicrobial effect of A. platensis on the bacteria was not confirmed. Additionally, please, see Lines 387-388: The phrase in the Discussion “These results suggest that A. platensis may play a crucial role in reducing the propagation of antibiotic resistance genes (ARGs) was not really conformed by results due to the absence of control without additions of microalgae to the wastewater.

Figure 4. I recommend to divide the long Table into two parts and show them in one Figure one under another by vertical position. Now the information at this Figure is unreadable.

Discussion

Please, add the information about antibiotic(s) present in the investigated “antibiotic-contaminated wastewater” (Line 258). Was the antibiotic concentration known? This information is important, because the decrease in bacteria concentration and changes in the content  of bacterial community (Figure 4) could be due to the presence of certain antibiotic in the VHW in a definite concentration for 8 days (period of wastewater exposition), but not due to the presence of A. platensis. There is not enough information in the article.

References

DOI should be added to all references where it is possible.

Author Response

Response to Reviewers

We would like to express our sincere gratitude to the reviewers for their thorough evaluation and constructive feedback on our manuscript. Their insightful comments and suggestions have been invaluable in improving the overall quality, clarity, and scientific rigor of the revised version. In the following section, we provide detailed, point-by-point responses to all comments, and we trust that our revisions adequately address the reviewers’ concerns.

General comment

1. The abbreviation VHW is introduced to the text several time (Lines 25, 165, 187, 193, 199, 227, 263). The situation should be avoided. The authors should use this abbreviation in the whole text (please, check the Lines 114, 149, 161, 162, 396, 407).

Authors response: We have replaced “veterinary hospital wastewater” with the abbreviation “VHW” consistently throughout the manuscript.

Abstract

2. Lines 30-31: The phrase “ platensis was cultivated for 8 days in Zarrouk medium supplemented with 0–100% VHW.” is not clear. Same situation is in Lines 114-115: “Zarrouk medium diluted with veterinary hospital wastewater at 0%, 25%, 50%, 75%, and 100% concentrations (v/v). “ Was there replacement of Zarrouk medium by the VHW per 0-100%? Replacement but not dilution? I suggest that the authors talk about diluting VHW using Zarrouk medium, since in many places in the text the authors use “25%VHW".

Authors response: We have revised the sentence for clarity by replacing the term “supplement” with “mixed”. The experimental setup is now described as follows: Zarrouk medium was mixed with non-sterile VHW at different ratios, 0% VHW (0 mL VHW + 100 mL Zarrouk medium), 25% VHW (25 mL VHW + 75 mL medium), 50% VHW (50 mL VHW + 50 mL medium), 75% VHW (75 mL VHW + 25 mL medium), and 100% VHW (100 mL VHW + 0 mL medium) (Lines 117–121).

3. Lines 39-40 (Abstract) and Lines 83-84 (in Introduction): Text of abstract (actual results obtained) on significant suppression of Proteobacteria cell growth and the motivation presented in the Introduction, according to which “the biological role of microalgal-bacterial symbiosis is crucial for the establishment and stability of the symbiotic system" contradict each other. There is a negative symbiosis for bacteria which was revealed by the authors in this work. In this regard, I recommend that the authors redo the text in the Introduction (Lines 75-87). It is necessary to focus on those studies that have already shown in other works that in addition to the possibility of accumulation of microalgae biomass containing pigments and reduction of nitrogen- and carbon-containing pollutants in wastewater, there is a significant antimicrobial effect of microalgae on bacterial cells present in wastewater.

Author Response: We have revised the Introduction to clarify microalgal-bacterial interactions. These interactions are diverse and complex, ranging from mutualistic symbiosis to competitive antagonism, and may also include commensalism and amensalism. In MBC systems, antagonistic effects can arise when decaying microalgae consumes oxygen or releases inhibitory compounds, limiting oxygen availability for bacteria. Additionally, under mixotrophic conditions, microalgal consumption of organic matter can further reduce net oxygen production, which restricts bacterial activity. Conversely, under a symbiotic rela-tionship, microalgae supply oxygen through photosynthesis to support bacterial nutrient utilization, while bacteria release carbon dioxide that microalgae utilize. Effective system performance depends on balancing these competing and cooperative dynamics through careful control of light, nutrient levels, and aeration. (Line 76–85).

Methods

4. Lines 106-111: Please, add the information about equipment used to realize all analytical methods in this work for the determination of ammonia nitrogen (NH₃-N), nitrile nitrogen, nitrate nitrogen (NO₃⁻-N), total phosphate (PO₄³⁻), biochemical oxygen demand (BOD), chemical oxygen demand (COD), total dissolved solids (TDS), total suspended solids (TSS), total Kjeldahl nitrogen (TKN), fats, oils, grease (FOG) and pH.

Author Response: We have added details on the analytical methods used, noting that all parameters were determined following the Standard Methods for the Examination of Water and Wastewater (APHA/AWWA/WEF) by Central Laboratory (Thailand) Company Limited (Lines 112-114).

5. Line 119-120: Please, add the equipment (type of photobioreactor) used for cell cultivation, intensity and regime (day/night or constant) of illumination.

Authors response: We have revised the sentence to specify the cultivation conditions. Cultures were incubated for 8 days on a rotary shaker at 120 rpm, under continuous LED illumination of 40 µmol/m²/s, at 30 ± 2 °C (Lines 126-127). A photobioreactor was not used; instead, cells were cultivated in flasks on a rotary shaker.

6. Line 126: Please add the type of centrifuge.

Authors response: We have specified the centrifuge model in the revised sentence (Line 134-135).

7. Lines 128: Please add the equipment used for measurements of absorbance at 665, 470, and 720 nm.

Authors response: We have specified the spectrophotometer model used to measure absorbance in the revised sentence. The absorbance of the extract was measured at 665, 470, and 720 nm using a Genesys 30 spectrophotometer (Thermo Fisher Scientific, USA) (Lines 137).

8. Please, the information about the methods of statistical treatment of obtained results in this work. It is important because there are following results in the text (Lines 171-173): “ 0.65 ± 0.05 g/L in the 50% VHW condition, and 0.58 ± 0.04 g/L) in control. Biomass production declined under the 75% VHW condition, resulting in a final concentration of 0.50 ± 0.05 g/L.” According to ANOVA there is no significant difference between results obtained for 50% VHW and Control, Control and 75% VHW…Please, see.

Authors response: We have revised the paragraph to correctly present the biomass results along with statistical analysis.

On day 8, biomass concentrations were 0.58 ± 0.04 g/L (0% VHW, control), 0.78 ± 0.05 g/L (25% VHW), 0.65 ± 0.05 g/L (50% VHW), and 0.50 ± 0.05 g/L (75% VHW). Biomass in the 25% VHW treatment was significantly higher than in the control (0% VHW) (p<0.05), while the 50% and 75% VHW treatments did not differ significantly from the control. In contrast, the 100% VHW condition significantly inhibited growth, with marked sup-pression observed during the first two days and complete growth cessation by day 4 (Fig-ure 1b).

We have also added details of the statistical analysis in Section 2, Materials and Methods. All data obtained in this study represent the means of three independent biological replicates, and the error bars represent the standard deviation (Mean ± SD, n = 3). The statistical analysis was analyzed by one way analysis of variance (ANOVA) and the significant difference (p < 0.05) were compared by Tukey’s HSD using SPSS version 22 (IBM, USA).

Results

9. Lines 152-159: The following text “Ammonia nitrogen and total Kjeldahl nitrogen (TKN) were found at concentrations of 56.56 ± 3.23 mg/L and 76.13 ± 3.10 mg/L, respectively, while nitrate nitrogen levels were very low (≤ 0.011 mg/L). Nitrile nitrogen was not detected. The samples also contained notable levels of organic pollutants, with a biochemical oxygen demand (BOD) of 71.50 ± 12.31 mg/L and a chemical oxygen demand (COD) of 160.97 ± 24.44 mg/L. Total phosphate was measured at 6.31 ± 1.43 mg/L. Regarding solids content, total dissolved solids (TDS) and total suspended solids (TSS) were recorded at 412.00 ± 21.57 mg/L and 25.06 ± 1.83 mg/L, respectively. Additionally, fat, oil, and grease were present at a concentration of 9.59 ± 2.58 mg/L.” should be considerable redone, because it contains same information which is performed in the Table 1. So, there is obvious doubling of same information in the article. The numbers shown in Table 1 should be deleted from the text.

Authors response: We have deleted the redundant information and revised the sentence.

10. Lines164-167: The text “ platensis was cultivated in 50 mL Zarrouk medium and in Zarrouk medium supplemented with non-sterile veterinary hospital wastewater (VHW) at varying ratios: 25%, 165 50%, 75%, and 100% (v/v). Cultures were maintained under continuous illumination (40 μmol/m²/s) for 8 days, with biomass and pigment contents recorded every two days.” repeats the text performed in section 2.2 (Methods), and here the authors mentioned cultivation for 8 days, whereas 7 days they gave in Line 118 (contradiction!).

Authors response: We have removed the sentence to avoid repetition of methodological details and have revised the cultivation period to 8 days, as suggested.

11. Figure 1(a) and figure capture: Please, check “Growth curves based on optical density at 730 nm (OD730)”. Are the authors sure that they controlled the growth of cells but not the chlorophyll-a accumulation at such wave length? Please, be correct and give the reference with recommendation of use such OD730 to control the cell biomass but not a fluorescent pigment. Please, see the Lines 128 and 130: there is the text with mentioning of 720 nm, but not 730 nm, and the extracts (!) from cells are discussed but not cell suspensions for estimation of biomass.

Authors response: The absorption spectrum of chlorophyll-a shows maxima at approximately 430 and 680 nm, while the region between 720–750 nm is considered an “absorbance minimum” where the influence of pigment content is negligible (Griffiths et al., 2011). Therefore, the choice of 730 nm, which lies close to this minimum region, mainly reflects light scattering associated with cell number rather than pigment absorption. In other words, OD730 serves as a direct indicator of cell biomass or cell density rather than chlorophyll-a accumulation. While OD730 was used to directly monitor cell biomass in intact cultures, chlorophyll content was determined separately from cell extracts. Specifically, a 1 mL culture sample was centrifuged and the pellet extracted with 99.9% methanol in the dark, after which the absorbance of the extract was measured at 665, 470, and 720 nm. Thus, the absorbance at 720 nm in this procedure served as a correction factor for background turbidity in the pigment extract and was not a direct measurement from intact cell suspensions.

We have also revised the manuscript by adding a description of the wavelength chosen to minimize interference from pigments (Lines 130).

(Griffiths, M.J.; Garcin, C.; van Hille, R.P.; Harrison, S.T.L. Interference by pigment in the estimation of microalgal biomass concentration by optical density. J Microbiol Methods. 2011, 85, 119–123.

https://doi.org/10.1016/j.mimet.2011.02.005.).

12. Figure 2. Please, see that the concentration of Chlorophyll-a (µg/mL) is 1,000 times less as compared to carotenoids (mg/mL)! It is impossible, because platensis cells have green color, but not red which is typical for carotenoids. Some unreal results are performed here. To check the situation, please see any other paper with the same cells. For example, this: DOI: 10.1039/C9GC03292D (Carotenoids, chlorophylls and phycocyanin from Spirulina: supercritical CO2 and water extraction methods for added value products cascade. Green Chem., 2020, 22, 187-196.).

Authors response: After careful verification, we found that the carotenoid concentration units were mistakenly reported as mg/mL, whereas they should be µg/mL. Correcting this unit resolves the apparent discrepancy, as it aligns with the expected relative concentrations of chlorophyll-a and carotenoids in A. platensis cells. We agree with the reviewer’s observation and have corrected the units accordingly.

13. Figure 3. There is NO very important control: the results of cultivation of 25%VHW with microorganisms initially present in this sample for 8 days without platensis! The pH of the medium used for cultivation was 10.0 ± 0.5 (see Line 119). This pH value is not convenient for the bacterial cells, so the action of the medium but not the presence of A. platensis for 8 days without fresh substrates for the bacteria could result in the decrease of their amounts. In fact, the antimicrobial effect of A. platensis on the bacteria was not confirmed. Additionally, please, see Lines 387-388: The phrase in the Discussion “These results suggest that A. platensis may play a crucial role in reducing the propagation of antibiotic resistance genes (ARGs) was not really conformed by results due to the absence of control without additions of microalgae to the wastewater.

Authors response: We have added additional shotgun analysis results, including raw wastewater for comparison. Three sample types were examined: 100% VHW (raw VHW), 25% VHW cultured without A. platensis (T25), and 25% VHW cultured with A. platensis (T25AP). Taxonomic composition was analyzed at the phylum, order, and genus levels. The analysis showed that some bacterial groups disappeared, which is likely due to the alkaline pH (~10) and high salinity of the Zarrouk-based medium rather than the presence of A. platensis. The medium imposes osmotic stress and limits bacterial growth because it relies on inorganic carbon sources such as NaHCO₃ and lacks the organic carbon required by many species. Exceptions include alkaliphilic or halotolerant bacteria, which can still grow under these conditions. In addition, co-cultivation with A. platensis (T25AP) further reshaped the microbial community, confirming that the observed microbial reductions were indeed due to the presence of A. platensis.

We have removed the paragraph suggesting that A. platensis may reduce the propagation of antibiotic resistance genes (ARGs) was not fully supported, due to the absence of a control without microalgae in the experiment.

14. Figure 4. I recommend to divide the long Table into two parts and show them in one Figure one under another by vertical position. Now the information at this Figure is unreadable.

Authors response: We have divided the long table into two parts and arranged them vertically in a single figure, one above the other, as suggested.

Discussion

15. Please, add the information about antibiotic(s) present in the investigated “antibiotic-contaminated wastewater” (Line 258). Was the antibiotic concentration known? This information is important, because the decrease in bacteria concentration and changes in the content of bacterial community (Figure 4) could be due to the presence of certain antibiotic in the VHW in a definite concentration for 8 days (period of wastewater exposition), but not due to the presence of platensis. There is not enough information in the article.

Authors response: We have revised the manuscript (Introduction and Discussion) to include the concentration range of antibiotics detected in VHW, as reported in a previous study, and have cited the appropriate references to support this revision (Lines 50–54 and 392–401).

While we did not perform quantification of antibiotic concentrations in this study, co-cultivation with A. platensis (T25AP) further reshaped the microbial community, confirming that the observed reductions in microbial abundance were directly attributable to the presence of A. platensis.

References

16. DOI should be added to all references where it is possible.

Authors response: We have added the DOI numbers to all references where available, as suggested.

Round 2

Reviewer 2 Report

Comments and Suggestions for Authors

Sorry, but I have the following commnets to the revised version of the article:

English language should be improved in the article.

The information about used equipment should be checked and the city ad state should be added to names of companies and countries. Please see lines 135, 137,145, etc.

Line 142: there is a phrase “cells were extracted in phosphate buffer”. The extraction as a process is typical for the compounds, but not cells. Probably, the term “suspended” is more appropriate here.

The equations in Section 2.3 are performed without their explanations. All of them were simply taken from other papers without understanding of meaning of the equations’ components. All components should be (including used coefficients!) disclosed in the text with the methods. This is very serious moment for the paper acceptance, because the results were obtained by using strange equations for the calculations. For example, there is the following text:” Biomass concentration was estimated spectrophotometrically by measuring the optical density at 730 nm”, but further we can read that “The absorbance of the extract was measured at 665, 470, and 720 nm”. Why was the length of the wave changed from 730 to 720 nm? What do the coefficients in the equations given in the article mean without numbering them?

Chl-a (µg/mL) = 12.9447 × (A₆₆₅ – A₇₂₀) 139

Car (µg/mL) = [1000 × (A₄₇₀ – A₇₂₀) – 2.86 × Chl-a] / 221

PC (mg/mL) = [A620 – (0.474 × A652)] / 5.34

Alp (mg/mL) = [A652 – (0.208 × A620)] / 5.09

There is a phrase from authors’ response “Thus, the absorbance at 720 nm in this procedure served as a correction factor for background turbidity in the pigment extract and was not a direct measurement from intact cell suspensions.”  Based on this information, please explain, why did the correction factor is used for the calculation of concentrations of chlorophyll (665 nm) and carotenoids (470 nm)and was not used for the same calculations of both phycocyanin (PC) and allophycocyanin (Alp) concentrations, and was no used for two other pigments, whereas the wavelength 720 nm is very close to 652 nm used for estimation of allophycocyanin?

Figures 3 C and D: I recommend to change the scale used for ordinate axis and put µg/mL instead of mg/mL. A lot of zeros will go away, and all pigments will be in comparable concentrations. Please, use same units for all pigments’ concentrations in the Abstract (lines 36-37).

Comments on the Quality of English Language

English language should be improved in the article.

Author Response

Dear Editor,

We sincerely thank the reviewers for their careful evaluation and constructive feedback on our manuscript. Their insightful comments have greatly contributed to enhancing the clarity, quality, and scientific rigor of the revised version. In the following section, we provide detailed, point-by-point responses to all comments, and we believe that our revisions effectively address the reviewers’ concerns.

Reviewer’ s comments

1. English language should be improved in the article.

Authors response: We have improved the English language throughout the article, as suggested.

2. The information about used equipment should be checked and the city and state should be added to names of companies and countries. Please see lines 135, 137,145, etc.

Authors response: We have carefully checked the information on the equipment and have added the city and state details to the names of the companies and countries, as suggested (Lines 128, 137, 140, 148, 153, 157, 158, 159, 160).

3. Line 142: there is a phrase “cells were extracted in phosphate buffer”. The extraction as a process is typical for the compounds, but not cells. Probably, the term “suspended” is more appropriate here.

Authors response: We appreciate the reviewer’s suggestion. We have revised the phrase accordingly, and now it reads “cells were suspended in phosphate buffer” (Line 144).

4. The equations in Section 2.3 are performed without their explanations. All of them were simply taken from other papers without understanding of meaning of the equations’ components. All components should be (including used coefficients!) disclosed in the text with the methods. This is very serious moment for the paper acceptance, because the results were obtained by using strange equations for the calculations. For example, there is the following text:” Biomass concentration was estimated spectrophotometrically by measuring the optical density at 730 nm”, but further we can read that “The absorbance of the extract was measured at 665, 470, and 720 nm”. Why was the length of the wave changed from 730 to 720 nm?

Author response: We sincerely thank the reviewer for raising this important point. In the revised manuscript, we have carefully explained each equation used in Section 2.3, including all components and coefficients, and provided the appropriate references for clarity. Regarding the wavelengths, we would like to clarify that absorbance at 730 nm was used for biomass concentration measurement directly from cell suspensions, while absorbance at 665 nm, 470 nm, and 720 nm was used for pigment analysis from solvent extracts. The difference arises from the distinct methodologies applied for biomass estimation versus pigment quantification. This has now been clearly explained in the Methods section to avoid any confusion (Section 2.3).

5. What do the coefficients in the equations given in the article mean without numbering them?

Chl-a (µg/mL) = 12.9447 × (A₆₆₅ – A₇₂₀) 139

Car (µg/mL) = [1000 × (A₄₇₀ – A₇₂₀) – 2.86 × Chl-a] / 221

PC (mg/mL) = [A620 – (0.474 × A652)] / 5.34

Alp (mg/mL) = [A652 – (0.208 × A620)] / 5.09

Author response: We appreciate the reviewer’s comment. In the revised manuscript, we have numbered all equations and clearly explained the meaning of each component used in the calculations, including references to the sources where the equations were reported. This clarification has been added to Section 2.3.

6. There is a phrase from authors’ response “Thus, the absorbance at 720 nm in this procedure served as a correction factor for background turbidity in the pigment extract and was not a direct measurement from intact cell suspensions.” Based on this information, please explain, why did the correction factor is used for the calculation of concentrations of chlorophyll (665 nm) and carotenoids (470 nm) and was not used for the same calculations of both phycocyanin (PC) and allophycocyanin (APC) concentrations, and was no used for two other pigments, whereas the wavelength 720 nm is very close to 652 nm used for estimation of allophycocyanin?

Author response: We appreciate the reviewer’s insightful comment. Chlorophyll absorbance is typically measured at ~665 nm. However, pigment extracts often contain suspended particles, cell debris, or other non-pigment compounds that cause light scattering and nonspecific absorption across the visible spectrum. This effect is most pronounced at longer wavelengths (near-infrared), where pigments themselves do not strongly absorb. Measuring the optical density (OD) at 720 nm provides a baseline estimate of this background turbidity. Therefore, zeroing the spectra at 720 nm is essential to minimize artifacts caused by light scattering from contamination (Chazaux, et.al., 2022)

Subtracting OD720 from the absorbance at pigment-specific wavelengths (e.g., 665 nm for chlorophyll, 470 nm for carotenoids) corrects for nonspecific contributions and yields a more accurate estimate of true pigment absorbance.

In contrast, pigments like phycocyanin and allophycocyanin have sharp, well-resolved peaks in regions less affected by scattering, so OD720 subtraction has little effect on their measurement. It is worth noting that studies on Arthrospira platensis measuring phycocyanin and allophycocyanin generally did not apply OD720 subtraction. For example, Khandual et al. (2021) and Tavanandi et al. (2018) performed their analyses without this correction. In contrast, Zavřel et al. (2018) applied OD720 subtraction when determining phycobiliprotein content in Synechocystis.

References

Chazaux, M.; Schiphorst, C.; Lazzari, G.; Caffarri, S. Precise estimation of chlorophyll a, b and carotenoid content by deconvolution of the absorption spectrum and new simultaneous equations for Chl determination. Plant J. 2022, 109, 1630-1648.

Khandual, S.; Sanchez, E.O.L.; Andrews, H.E.; de la Rosa, J.D.P. Phycocyanin content and nutritional profile of Arthrospira platensis from Mexico: efficient extraction process and stability evaluation of phycocyanin. BMC Chem. 2021, 15, 24.

Tavanandi, H.A.; Chandralekha Devi, A.; Raghavarao, K.S.M.S. A newer approach for the primary extraction of allophycocyanin with high purity and yield from dry biomass of Arthrospira platensis. Sep. Purif. Technol. 2018, 204, 162–174.

Zavřel, T.; Chmelík, D.; Sinetova, M.A.; Červený, J. Spectrophotometric determination of phycobiliprotein content in cyanobacterium Synechocystis. J. Vis. Exp. 2018, 11(139), 58076.

7. Figures 2C and D: I recommend to change the scale used for ordinate axis and put µg/mL instead of mg/mL. A lot of zeros will go away, and all pigments will be in comparable concentrations.

Authors response: We have revised Figures 2C and 2D by changing the ordinate axis scale from mg/mL to µg/mL, as recommended. This adjustment removes unnecessary zeros and allows for better comparison of pigment concentrations.

8. Please, use same units for all pigments’ concentrations in the Abstract (lines 36-37).

Authors response: We have corrected the units and ensured consistency for all pigment concentrations in the Abstract (Lines 36-37, 192), as suggested.